# Anatomy of a Hybrid Mind: Deconstructing Hybrid Reasoning in Large Language Models

## Abstract

Hybrid reasoning, which enables a *single* Large Language Model (LLM) to alternate between fast, intuitive responses (non-thinking mode) and slow, deliberate reasoning (thinking mode), has rapidly gained adoption across the AI industry, spanning from top-tier commercial models to the latest open-source releases. Despite their widespread deployment, the community still lacks a mechanistic understanding of how these two modes coexist and interact *within a single model*. Without such clarity, we lack a principled understanding of how these modes operate, making it harder to reason about their behavior or guide future development. In this work, we conduct a detailed mechanistic analysis of a hybrid reasoning model's internal dynamics. We identify why the thinking and non-thinking modes can be compatible rather than distinct subsystems. Building on this, we propose a metacognitive taxonomy showing that the thinking mode corresponds to a structured, self-corrective protocol whose intensity can be modulated along a continuous spectrum. Furthermore, we causally uncover a surprisingly localized, single-token switch that deterministically governs mode activation. These findings illuminate the control mechanisms underlying hybrid reasoning, providing a foundation for the design of robust, interpretable, and adaptive cognitive systems.

## 1 Introduction

The pursuit of artificial general intelligence has driven Large Language Models (LLMs) to achieve impressive capabilities in complex reasoning. The technique of Chain-of-Thought (CoT) prompting (Wei et al., 2022), which encourages models to "think step by step", has shown that performance on challenging tasks can be significantly improved through explicit and deliberate reasoning. While this thinking mode is powerful, its lengthy process introduces substantial latency, proving unnecessary for simpler queries where users expect fast and intuitive responses. Initially, this dilemma was addressed by deploying two distinct models: a powerful thinking model optimized for long-form reasoning (*e.g.*, OpenAI o1 (Jaech et al., 2024), DeepSeek-R1 (Guo et al., 2025)) and a swift non-thinking model for instant answers (*e.g.*, OpenAI GPT-4o (Hurst et al., 2024), DeepSeek-V3 (Liu et al., 2024)).

More recently, as shown in Figure 1, the concept of *hybrid reasoning* has emerged as an elegant solution, aiming to integrate these two distinct modes into a single, unified architecture (Anthropic, 2025a). This approach has been rapidly adopted, with major organizations releasing their own hybrid reasoning models, from the closed-source Claude 4 (Anthropic, 2025b) to open-source alternatives like Qwen3 (Yang et al., 2025) and Llama-Nemotron (Bercovich et al., 2025).

Despite its wide adoption, we still lack a clear understanding of how hybrid reasoning actually works. This gap in our knowl-

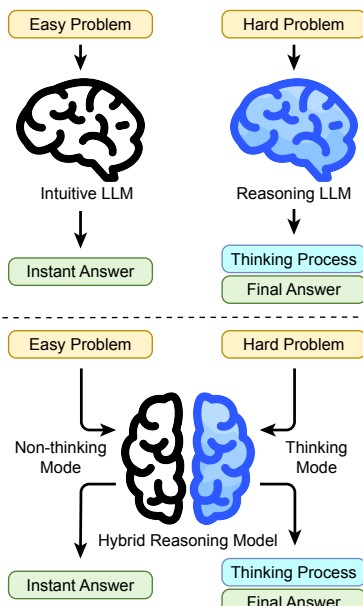

Figure 1: A hybrid reasoning model integrates the thinking and non-thinking modes into a single, unified architecture.

edge raises several fundamental questions. **Q1:** Why are the two modes compatible within a single model rather than functioning as conflicting subsystems? **Q2:** When one mode is selected, what exactly is being activated that leads to such striking differences in behavior? **Q3:** How is this activation controlled? Is it a complex and distributed process, or a simple and localized switch? Without answers, the community risks building on unstable foundations: hybrid reasoning may remain a powerful but opaque trick, rather than a principled paradigm for LLM reasoning.

In this paper, we take the first step towards closing this gap through a detailed mechanistic study of hybrid reasoning models. Specifically, we focus on models where the reasoning mode is manually controlled, typically toggling between thinking and non-thinking (though some variants offer multiple levels, such as low, medium, and high). This setting provides a controlled environment to investigate the nature of these modes and the mechanisms governing them. Based on this setting, we investigate the questions we mentioned in the last paragraph and present a deep anatomical study into the internal workings of hybrid reasoning. By treating a state-of-the-art model as a case study, we employ a series of targeted experiments, from probabilistic analysis to causal interventions, to dissect the relationship, control, and nature of its reasoning modes. Our contributions are as follows:

- **We explain why the two modes are compatible (Q1).** Our cross-evaluation reveals that the thinking and non-thinking modes mirror the relationship between reasoning and base models: in both cases, divergence occurs only on a small subset of tokens. This minor difference explains the good compatibility of the two modes.

- **We define the essence of "thinking" as a structured metacognitive protocol (Q2).** We propose a novel taxonomy of metacognitive keywords and show quantitatively that the thinking mode is characterized by the activation of a protocol involving planning, self-correction, reasoning articulation, and deliberation.

- **We show that reasoning intensity is a controllable spectrum (Q2).** By extending our analysis to a model with multiple reasoning levels, we reveal that this metacognitive activation is not only a binary switch but a modulatable spectrum, allowing for fine-grained control over the intensity of the reasoning process.

- **We identify a localized, single-token control mechanism (Q3).** We use probabilistic analysis to locate a single token as the likely control switch and provide causal evidence through forced decoding experiments, demonstrating that this token deterministically activates or deactivates the model's deliberative reasoning process.

By systematically answering these questions, our work moves beyond treating reasoning modes as black-box commands and offers a clear, mechanistic account of how "thinking" is implemented and controlled. This foundational understanding is a crucial first step towards building the robust hybrid reasoning models of the future.

## 2 PRELIMINARIES AND RELATED WORK

### 2.1 REASONING MODES IN LLMS

Previous LLMs can broadly be divided into two categories. *Reasoning LLMs* leverage techniques such as chain-of-thought prompting (Wei et al., 2022) to generate explicit, step-by-step intermediate reasoning. These models often achieve higher accuracy on complex tasks but induce substantial latency due to reasoning and lengthy outputs. In contrast, *intuitive LLMs* produce direct answers without intermediate steps, enabling faster interaction but often with a limitation in reasoning depth.

Hybrid reasoning is proposed to unify these two paradigms within a single model. Anthropic's Claude 3.7 Sonnet (Anthropic, 2025a) is the first hybrid reasoning model on the market, with fast answers for simple prompts and extended reasoning for complex ones. Such ability is also inherited by the subsequent Claude 4 (Anthropic, 2025b). In the open-source community, Alibaba's Qwen3 (Yang et al., 2025) and NVIDIA's Llama-Nemotron (Bercovich et al., 2025) are released around the same time as the first open-source model series supporting hybrid reasoning, followed by Zhipu AI's GLM-4.5 (Zeng et al., 2025a). More recently, Kuaishou's KAT-V1 (Zhan et al., 2025) introduces an *AutoThink* mechanism that adaptively decides whether to engage in step-by-step reasoning. The gpt-oss series (Agarwal et al., 2025) is a variant of these hybrid reasoning models, which supports three different reasoning efforts: low, medium, and high.

## 2.2 Manual Control in Hybrid Reasoning Models

Current hybrid reasoning models expose these modes to users through explicit control mechanisms. In particular, all major open-source hybrid reasoning models adopt a *prompt-based control* strategy, utilizing the insertion of a *specific token* into the prompt to tell the model whether to switch into deliberative reasoning (Yang et al., 2025; Bercovich et al., 2025; Zeng et al., 2025a; Zhan et al., 2025; Agarwal et al., 2025). Some models further extend this paradigm by allowing multiple reasoning levels (*e.g.*, low, medium, high), each triggered by a distinct control token. For instance, Qwen3 (Yang et al., 2025) uses `</think>` and `</nothink>` to toggle between modes; Llama-Nemotron (Bercovich et al., 2025) sets a `detailed thinking on/off` flag in the system prompt; GLM-4.5 (Zeng et al., 2025a) defaults to the thinking mode unless `</nothink>` is given; and KAT-V1 (Zhan et al., 2025) employs `<think_on>` and `<think_off>`. The gpt-oss series (Agarwal et al., 2025) instead allows users to specify different reasoning efforts (low, medium, high) directly in the system prompt. This design provides a controlled experimental setting to analyze how reasoning modes are implemented internally, since the behavior of the model can be toggled deterministically through prompt intervention.

## 3 On the Compatibility of Reasoning Modes

Before analyzing the specific mechanism underlying a hybrid reasoning model, we must first address a more fundamental question: why are these seemingly disparate modes, one optimized for fast and intuitive responses and the other for slow and deliberate reasoning, compatible within a single model? From an intuitive perspective, given the significant differences in behavior between the two modes, merging them seems highly challenging. In this section, we challenge this intuition. We argue that the actual behavioral divergence between a base model and its specialized version is surprisingly small, and this low divergence is the fundamental reason why they are compatible.

To empirically test this claim, we first quantify the divergence between base LLMs and their specialized, fine-tuned versions. These model pairs represent a real-world scenario where a new, specialized capability (*e.g.*, long-form mathematical reasoning) is successfully integrated into an existing model. By measuring their difference, we can directly assess the true extent of behavioral change that specialization entails. We analyze two distinct pairs to ensure the breadth of our findings across different fine-tuning paradigms:

- **Supervised Fine-Tuning (SFT):** We compare a base model, Qwen2.5-Math-7B (Yang et al., 2024), with its SFT derivative, DeepSeek-R1-Distill-Qwen-7B (Guo et al., 2025), which is distilled from reasoning data generated by DeepSeek-R1 and is specialized for reasoning.

- **Reinforcement Learning (RL):** We compare the base Qwen2.5-Math-7B model with its derivative, Qwen-2.5-Math-7B-SimpleRL-Zoo (Zeng et al., 2025b), which has been fine-tuned for higher reasoning abilities using RL.

To measure the mutual divergence between two models in a pair (*i.e.*, a base model and its fine-tuned version), we conduct a symmetric cross-evaluation as shown in Algorithm 1. For a given set of prompts, we generate responses with each model and let the other model in the pair evaluate them. We call them the *generating model* and the *evaluating model* respectively. We quantify the divergence from these evaluations using two key metrics:

---

**Algorithm 1** Cross-evaluation

1: **Inputs:** Prompt $q$; *generating model* (mode) $A$; *evaluating model* (mode) $B$.
2: **Outputs:** $B$'s predicted probabilities of each token in $A$'s output.
3: $a_A \leftarrow \text{Generate}(A, q)$
4: $p_{B \leftarrow A} \leftarrow \text{Evaluate}(B, q + a_A)$
5: **return** $p_{B \leftarrow A}$

---

- **Average Negative Log-Likelihood (NLL):** This metric quantifies how "surprised" an *evaluating model* is by the output sequence from a *generating model*. For a response $Y = (y_1, \ldots, y_T)$, we calculate the average NLL of each token according to the *evaluating model*'s predicted probability: $\text{NLL} = -\frac{1}{T} \sum_{t=1}^{T} \log P_{M_{\text{eval}}}(y_t | y_{<t})$. A lower NLL signifies higher plausibility and thus lower divergence.

- **Top-1 Prediction Agreement:** This is an intuitive measure of behavioral alignment. It calculates the percentage of tokens where the *evaluating model's* top-predicted next token exactly matches the token actually produced by the *generating model*. The agreement is calculated as

Table 1: Divergence metrics for specialized models versus their base models. Despite extensive fine-tuning, the token-level top-1 disagreement remains below 10%, challenging the notion of a large behavioral gap and revealing the basis for compatibility.

| Generating Model | Evaluating Model | Avg. NLL | Top-1 Agreement (%) |
|---|---|---|---|
| DeepSeek-R1-Distill | Qwen2.5-Math | 1.2652 | 91.95% |
| Qwen2.5-Math | DeepSeek-R1-Distill | 1.4173 | 97.76% |
| SimpleRL-Zoo | Qwen2.5-Math | 1.1538 | 95.54% |
| Qwen2.5-Math | SimpleRL-Zoo | 1.2685 | 97.97% |

Table 2: Divergence metrics between the thinking and non-thinking modes of Qwen3-8B and Llama-3.1-Nemotron-Nano-8B-v1.

| Model | Generating Mode | Evaluating Mode | Avg. NLL | Top-1 Agreement (%) |
|---|---|---|---|---|
| Qwen3-8B | thinking | non-thinking | 1.1631 | 95.33% |
| | non-thinking | thinking | 1.1828 | 95.25% |
| Llama-Nemotron-8B | thinking | non-thinking | 1.3283 | 91.59% |
| | non-thinking | thinking | 1.2219 | 94.53% |

$\frac{1}{T} \sum_{t=1}^{T} \mathbb{I}(y_t = \hat{y}_t)$, where $\hat{y}_t$ is the most probable token predicted by the evaluating model at step $t$. A high agreement rate signals a strong alignment in their predictive patterns.

The results, presented in Table 1, offer a striking rebuttal to the common intuition of a large behavioral gap. Despite extensive specialization through both SFT and RL, the token-level agreement between the base and fine-tuned models remains consistently above 90%. This core finding demonstrates that significant functional specialization can be achieved with less than 10% token-level behavioral divergence. This underlying similarity is the primary reason why integrating these distinct modes is feasible: they are not as different as they seem, making their coexistence within a unified model practical.

Having established this principle of low divergence, we now conduct a corroborating analysis on the thinking and non-thinking modes of the hybrid reasoning models Qwen3-8B (Yang et al., 2025) and Llama-3.1-Nemotron-Nano-8B-v1 (Bercovich et al., 2025). This verifies that the internal modes of hybrid reasoning models share the same pattern of high behavioral similarity. As shown in Table 2, the divergence between modes is indeed small and comparable to specialized models. This confirms that current hybrid reasoning models are successful instances of the low-divergence compatibility principle, with their two modes coexisting harmoniously within a single model.

> **Takeaway 1:** Contrary to intuition, specialization via fine-tuning (SFT/RL) induces only a small ($<10\%$) token-level behavioral divergence from a base model. This low divergence is the key reason why a hybrid model's distinct reasoning modes can be compatibly integrated within a single architecture.

## 4 THE ESSENCE OF "THINKING": A MODULATABLE METACOGNITIVE PROTOCOL

In the previous section, we established that the token-level behavioral divergence between the thinking and non-thinking modes is quite small. This presents a crucial puzzle: if the divergence between the modes is so subtle, where precisely do these two modes differ in the generative process? And more importantly, how can such a seemingly insignificant difference account for the profound, qualitatively distinct outcomes we intuitively associate with "thinking"? To resolve this, we now shift our focus to dissecting the nature of the behaviors in the two modes. [1]

---

[1] Although Wang et al. (2025) find that a small part of tokens are special to the reasoning model, our findings are fundamentally different. Specifically, Wang et al. (2025) examine only the entropy of tokens within a single reasoning mode, whereas we compare the logits of tokens across two modes. Our conclusion clarifies the distinction between two modes, rather than describing the behavior of a single mode as in Wang et al. (2025).

## 4.1 FROM PROBABILISTIC DIVERGENCE TO THEMATIC PATTERNS

Our initial clue comes from the NLL analysis in Section 3, which reveals systematic probabilistic divergences between the modes. To understand the qualitative nature of these differences, we seek to identify the specific tokens and contexts that consistently cause one mode to be "surprised" by the other (*i.e.*, tokens with high NLL). To facilitate this investigation, we develop an interactive visualization tool that renders the token-level NLL in an explorable HTML format, which we include in our supplementary materials. An example of such visualization is shown in Figure 2.

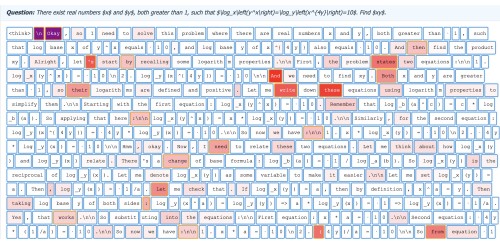 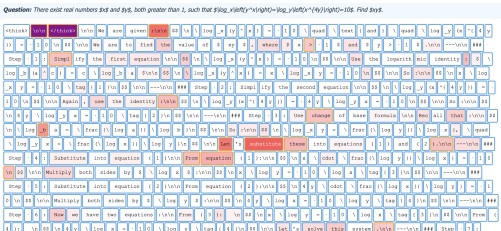

(a) Generating mode: thinking; evaluation mode: non-thinking.

(b) Generating mode: non-thinking; evaluation mode: thinking.

Figure 2: Two examples of rendering the token-level NLL. Redder tokens indicate higher NLL, while whiter tokens indicate lower NLL. Dark magenta tokens have NLL larger than 3. Blue borders show where the generated token matches the top-1 prediction of the *evaluating mode*, and orange borders show mismatches. More examples can be found in our supplementary HTML file.

By systematically examining a large and diverse set of samples through this tool, we move beyond aggregate statistics to a qualitative, data-driven analysis. We observe that the points of highest divergence are not random but consistently fall into several recurring thematic categories. For instance, the non-thinking mode assigns low probability to tokens like `Wait`, `Let's double-check`, or `Hold on` that frequently appear in the thinking mode's output. Conversely, the thinking mode is surprised by the non-thinking mode's tendency to jump directly into calculations or final answers. These observations strongly suggest that the core difference is not in the model's knowledge but in its application of strategic, metacognitive behaviors. We inductively grouped these observed behaviors into four primary functions, forming the basis of our taxonomy.

## 4.2 QUANTITATIVE DISSECTION BASED ON A TAXONOMY OF METACOGNITIVE KEYWORDS

Based on these bottom-up observations, we formalize our findings into a taxonomy of metacognitive keywords designed to quantify the reasoning style of an LLM. This taxonomy categorizes words and phrases into four distinct functions (the full list can be found in Appendix A):

**A. Planning & Structuring:** Words that signal the organization of the problem-solving process. These keywords establish a high-level plan or break the problem down into sequential steps. Examples: *Let's plan, First, Step 1, The overall approach is, Let's break it down.*

**B. Self-Correction & Verification:** Words that indicate reflective behavior, where the model pauses to check, validate, or correct its own reasoning or calculations. This is a hallmark of deliberate cognition. Examples: *Wait, Hold on, Let's check, Re-evaluate, Correction.*

**C. Reasoning Articulation:** Words that explicitly link steps in a logical chain, conveying the "why" behind a conclusion rather than just stating the "what". Examples: *Because, Therefore, This implies, The reason is, It follows that.*

**D. Deliberation & Uncertainty:** Words that express consideration of alternatives, assumptions, or confidence levels. This reflects a more cautious and nuanced approach to problem-solving. Examples: *Let's consider, Perhaps, Maybe, It seems that, Let's assume.*

Equipped with this taxonomy, we perform a large-scale analysis on the outputs of Qwen3-8B in both its thinking and non-thinking modes across our evaluation benchmarks. Figure 3 presents the frequency and absolute counts of keywords from each category on the AIME24 dataset (Li et al., 2024). For Self-Correction (B), Reasoning Articulation (C), and Deliberation (D), the thinking

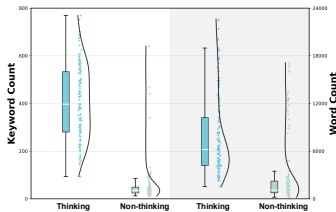 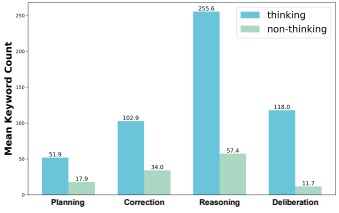 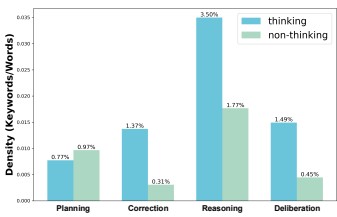

(a) The keyword count and the total word count of both modes.

(b) Comparison of mean keyword counts between the two modes.

(c) Comparison of the keyword densities between the two modes.

Figure 3: The statistics about the metacognitive keywords of Qwen3-8B on the AIME24 dataset. In Subfigure (a), box plots show the median (line), 25th–75th percentiles (box), and $1.5\times$ IQR (whiskers). On the right side of each box plot, points show the distributions and lines indicate the kernel density. The other two subfigures show the mean keyword counts and densities between the two modes. Although the thinking mode has more metacognitive keywords among all four types, its density of planning keywords is less than that of the non-thinking mode.

mode demonstrates a dramatically higher frequency and count of metacognitive keywords. This confirms that the "thinking" process is characterized by an active protocol of self-monitoring, logical explanation, and careful consideration. More nuanced is the finding for Planning & Structuring (A). While the thinking mode uses more planning keywords in absolute terms (due to longer, multi-step solutions), their *frequency* is lower than in the non-thinking mode. This reveals a key insight: the non-thinking mode is a "plan-executor" that concisely lists steps (*"Step 1... Step 2..."*), while the thinking mode is a "plan-debator" that spends significant generative budget between each planning step on verification and articulation, thereby lowering the density of planning words. More results on other datasets can be found in Appendix C.

### 4.3 METACOGNITIVE KEYWORDS AS CATALYSTS FOR DIVERGENCE

The identification of metacognitive keywords raises a central question: what is their relationship with the model's internal generative uncertainty? Our initial visual analysis in Section 4.1 suggests that high NLL tokens tend to appear immediately after these keywords. To test this, we conduct a reverse analysis: we first isolate the moments of greatest model surprise by selecting the top 5% of all tokens based on their NLL scores (NLL $> 0.8867$). For each of these high-NLL tokens, we then measure its positional relationship to the nearest metacognitive keyword.

The results are clear: 79.22% of all high-NLL tokens occur within 3 tokens of a metacognitive keyword. As illustrated in Figure 4, this tight coupling is not coincidental. It reveals the critical role of these keywords as "triggers" that are immediately followed by high divergence in the model's generative path. For instance, words like `Perhaps` or `Let's check` precede a spike in the NLL of subsequent tokens because they open up multiple possibilities for the next reasoning step. This experiment not only quantifies our initial visual findings but, more importantly, directly links the linguistic-level concept of metacognitive keywords to the probabilistic-level uncertainty within the model, thus forging a complete explanatory chain.

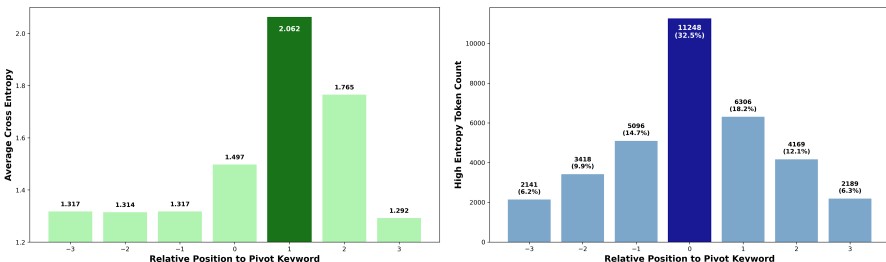

Figure 4: Average NLL of tokens around metacognitive keywords and their counts, categorized by the position to the pivot metacognitive keywords. Although the keywords themselves are the majority of the high NLL tokens, the ones right after them have higher average NLL scores.

## 4.4 GENERALIZATION TO A SPECTRUM OF COGNITION

Our finding that a metacognitive protocol can be toggled on or off is not limited to a binary switch. The same principle extends to models that exhibit a more granular spectrum of cognitive control. To demonstrate this, we analyze OpenAI's gpt-oss-20B (Agarwal et al., 2025), which supports three distinct reasoning efforts: low, medium, and high.

Applying our metacognitive taxonomy to the outputs of gpt-oss-20B reveals a clear, monotonic trend. The frequency of keywords related to Self-Correction (B), Reasoning Articulation (C), and Deliberation (D) systematically decreases from the high to the low setting, and the frequency of keywords related to Planning & Structuring (A) increases, similar to the observations on Qwen3-8B. This suggests that the underlying mechanism is not a simple on/off switch but rather a dimmer that modulates the degree of metacognitive engagement. The low mode closely resembles the concise, plan-executing behavior of Qwen3's non-thinking mode, whereas the high mode engages in a significantly more intensive and explicit reflective protocol.

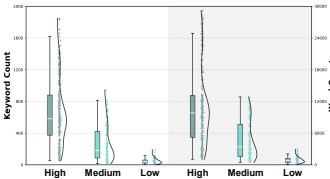 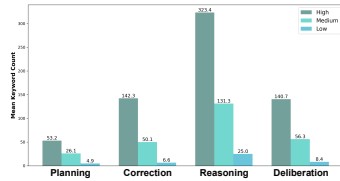 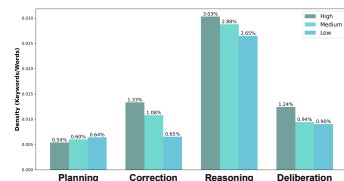

(a) The keyword count and the total word count of three modes.

(b) Comparison of mean keyword count among the modes.

(c) Comparison of the keyword density among the modes.

Figure 5: The statistics about the metacognitive keywords on gpt-oss-20B. Although the metacognitive keyword counts decrease from the high to the low setting among all four types, the density of planning keywords increases from the high to the low setting, which is similar to the phenomena observed on Qwen3-8B.

> **Takeaway 2:** The essence of the "thinking" mode lies not in enhanced core intelligence but in the activation of a structured metacognitive protocol characterized by planning, self-correction, reasoning articulation, and deliberation. Within this process, metacognitive keywords act as catalysts that trigger significant probabilistic divergence in the model's generative path, and models with more modes thereby reveal a more granular spectrum of cognitive control.

## 5 UNVEILING THE CONTROL MECHANISM: A LOCALIZED SWITCH

Having established that the thinking and non-thinking modes share similar behaviors (Section 3) and activate distinct metacognitive protocols (Section 4), a crucial question emerges: how does the model toggle between these behaviors? Our investigation begins with a striking observation in Figure 2: the token generated immediately after the `<think>` start tag consistently exhibits an abnormally high NLL during cross-evaluation. This suggests that this specific token position is a point of extreme disagreement between the two modes. In this section, we build on this initial clue, using probabilistic and causal analyses to demonstrate that the control mechanism is not a complex, distributed process but a remarkably simple, localized switch.

### 5.1 AN ANOMALOUS SPIKE IN PROBABILISTIC DIVERGENCE

To systematically quantify our initial observation, we plot the histogram of the token-level Kullback-Leibler (KL) divergence between the two modes' predicted probability distributions and NLL at each token position. These two metrics allow us to measure the "surprise" or disagreement between the modes' predictions across the entire generation process. As shown in Figure 6, the resulting histogram displays a typical long-tail distribution, confirming that for the vast majority of tokens, the two modes are in close agreement (KL/NLL values near zero). However, the plot also reveals

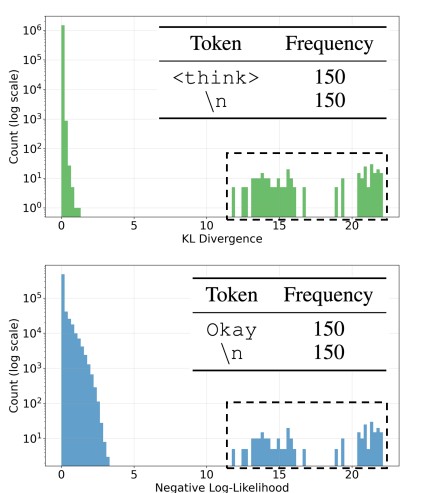
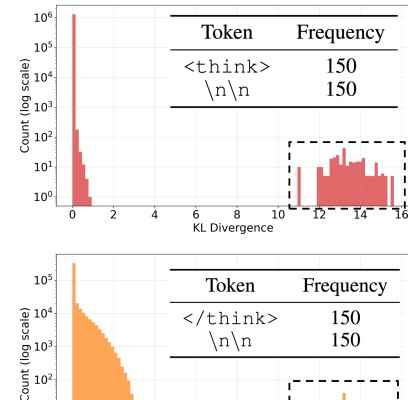

(a) Generating mode: thinking; evaluation mode: non-thinking.

(b) Generating mode: non-thinking; evaluation mode: thinking.

Figure 6: Distribution of token-level KL divergence and NLL between the thinking and non-thinking modes. In all four subfigures, anomalous spikes at high values can be found and are almost entirely attributed to the first few tokens following the `<think>` tag. For the thinking mode, the first three tokens generated are always `<think>\n Okay` and then followed by the thinking process. For the non-thinking mode, the first three tokens generated are always `<think>\n\n</think>`.

an isolated spike at the high-divergence end of the spectrum. This spike is too pronounced and localized to be explained by the long-tail phenomenon and points towards a systematic point of extreme disagreement.

By isolating the tokens that constitute this spike, we confirmed our initial finding. The spike is almost exclusively composed of the first few tokens generated immediately after the prompt. Specifically, for the thinking mode, the model overwhelmingly predicts `\n Okay` after the token `<think>`, while for the non-thinking mode, it predicts `\n\n</think>` (the closing tag of the thinking process). The extreme divergence at this specific position provides a powerful hypothesis: the choice of this token is not merely a stylistic artifact but could function as a localized switch that determines the model's entire subsequent reasoning trajectory.

## 5.2 CAUSAL VERIFICATION VIA FORCED DECODING

The probabilistic clue from the KL divergence analysis is a strong correlation, but it does not establish causality. To test the hypothesis that this single token acts as a causal switch, we designed a forced decoding experiment. In this intervention, we manually override the model's predicted first token after the `<think>` tag and observe whether its subsequent behavior is altered. We created two counterfactual scenarios:

1. **Deactivating the Think Mode:** We run the model with the `/think` command but force the first few tokens after `<think>` to be `\n\n</think>`, effectively mimicking the non-thinking mode's initial token.

2. **Activating the Think Mode:** Conversely, we run the model with the `/nothink` command (which internally expands to `<think>\n\n</think>`) but force the first two tokens after `<think>` to be `\n Okay`, the usual starter for the thinking mode.

The results, summarized in Table 3, provide decisive causal evidence. When the thinking mode is forced to generate `</think>`, its behavior flips entirely: the average output length and task performance become nearly identical to that of the genuine non-thinking mode. Conversely, forcing the non-thinking mode to start with `Okay` compels it to engage in a full-fledged, step-by-step reasoning process, with its output characteristics and performance aligning closely with the genuine thinking

Table 3: Results of the forced decoding experiments. Forcing the initial token causally flips the model's reasoning mode, aligning its performance and output length with the mode it is forced to imitate.

| Prompt | AIME24 | | AMC | | Math500 | | Minerva | | Olympic Bench | |
|---|---|---|---|---|---|---|---|---|---|---|
| | Acc | Len | Acc | Len | Acc | Len | Acc | Len | Acc | Len |
| Genuine `/think` | 78.00 | 14000.03 | 88.67 | 10496.04 | 96.00 | 5257.47 | 46.32 | 6920.73 | 68.41 | 11292.19 |
| Genuine `/nothink` | 25.33 | 9056.89 | 60.24 | 3493.67 | 84.84 | 1518.41 | 32.06 | 685.94 | 53.04 | 3366.26 |
| `/think` + Forced `</think>` | 29.33 | 8829.36 | 64.33 | 3635.75 | 86.80 | 1586.66 | 32.87 | 701.87 | 54.22 | 3737.83 |
| `/nothink` + Forced `Okay` | 76.00 | 15267.97 | 90.12 | 10413.30 | 96.12 | 5267.83 | 46.25 | 6256.37 | 68.80 | 10994.38 |

mode. This intervention causally demonstrates that the choice of the first token after the `<think>` tag is not only a stylistic artifact but serves as a deterministic switch that activates or deactivates the entire deliberative reasoning protocol.

> **Takeaway 3:** The control mechanism for switching between reasoning modes is not distributed but is highly localized to a single token position. Specifically, these single tokens following the control tag act as a causal, deterministic switch, capable of flipping the model's entire subsequent reasoning behavior.

## 6 DISCUSSION AND CONCLUSION

In this work, we deconstruct the mechanics of hybrid reasoning by systematically addressing three fundamental questions. First, we explain why the thinking and non-thinking modes are compatible within a single model rather than functioning as conflicting subsystems. We then uncover the nature of what is activated when the thinking mode is engaged, showing that it corresponds to a structured metacognitive protocol whose intensity can be modulated along a continuous spectrum. Finally, we identify how this activation is controlled, revealing a surprisingly localized, single-token switch that deterministically toggles the reasoning process. Together, these findings reframe hybrid reasoning as the controlled activation of a specific cognitive module, offering both a mechanistic explanation and a blueprint for building more efficient and controllable reasoning systems. Beyond a theoretical contribution, our findings illuminate several potential pathways for advancing the efficient training and deployment of reasoning models.

**Accelerating Reinforcement Learning for Reasoning.** A major bottleneck in improving the LLM's reasoning ability via RL is the high cost of sampling from the model during the policy roll-out. Our discovery shows that non-metacognitive reasoning steps are highly stable and exhibit low probabilistic variance, which creates favorable conditions for accelerating sampling with *retrieval-based speculative decoding* (He et al., 2024b; Hu et al., 2025). In this paradigm, long sequences of predictable tokens (*e.g.*, standard calculations or boilerplate phrases) can be cached from previous generations. During RL rollouts, these cached sequences can be retrieved and verified in parallel, while the main model is only invoked for high-variance, decisive tokens, which we identify as being concentrated around metacognitive keywords. This approach is particularly attractive as it obviates the need for training and maintaining a separate draft model, potentially streamlining the RL rollout process and reducing inference latency during serving.

**Efficient and Targeted Gradient-Based Learning.** The observation that token-level divergence between reasoning modes clusters around metacognitive keywords suggests a potential for more resource-efficient training strategies. One could explore a targeted gradient update scheme where backpropagation is focused primarily on the high-variance metacognitive tokens and their immediate context. The hypothesis is that the most critical learning signals for refining complex reasoning are concentrated in these strategic decision points. By applying gradients selectively, one might preserve a significant portion of the performance gains while substantially reducing the computational cost per training step. Such an efficiency gain would allow for greater training throughput, enabling the model to learn from a larger number of samples within the same time or computational budget, offering a compelling trade-off for developing sophisticated reasoning abilities at scale.

## REPRODUCIBILITY STATEMENT

We have taken multiple steps to ensure the reproducibility of our results. All experimental settings, including model configurations, prompting strategies, and evaluation protocols, are described in detail in Appendix B. The datasets used (primarily AIME) are publicly available, and we describe the sampling and evaluation methodology to allow consistent replication. Implementation details such as generation backends, hyperparameter settings, and evaluation libraries are documented in Appendix B.

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

## A   METACOGNITIVE KEYWORDS LIST

To support the taxonomy introduced in the main text, we provide the complete list of metacognitive keywords and phrases used in our analysis. These keywords are manually curated and grouped into four functional categories. The list is not intended to be exhaustive; rather, it captures the most frequently observed signals of metacognitive reasoning in LLM outputs. Researchers may use this list as a reference for annotation, quantitative analysis, or further refinement of the taxonomy.

Table 4: Comprehensive list of metacognitive keywords and phrases categorized into four functional groups. This list serves as the basis for our quantitative analysis in the main text (Section 4).

| Category | Keywords / Phrases |
|---|---|
| Planning & Structuring | plan, first, second, third, next, step, approach, solve, break, breakdown, structure, organize, method, strategy, procedure, process, sequence, order, phase, stage, begin, start, initial, finally, last, end; 
 *Phrases:* let's plan, the plan is, overall approach, to solve this, let's break it down, step by step, first step, next step, final step, in order to, the goal is |
| Self-Correction & Verification | wait, hold, check, double-check, verify, confirm, evaluate, reevaluate, reconsider, rethink, review, correction, correct, mistake, error, wrong, incorrect, actually, realize, notice, see, oops, sorry, revise, adjust, fix, amend, modify, update; 
 *Phrases:* hold on, let's check, let's double-check, let me check, let me verify, let me reconsider, let me rethink, that's wrong, that's incorrect, I made a mistake, I was wrong, actually no, wait no |
| Reasoning Articulation | because, therefore, hence, thus, so, since, consequently, accordingly, follow, imply, mean, reason, explain, why, how, what, where, when, result, conclude, conclusion, deduce, infer, logic, logical, reasoning, rationale; 
 *Phrases:* this implies, it follows that, the reason is, this means, in other words, that is to say, as a result, we can conclude, this shows |
| Deliberation & Uncertainty | consider, think, suppose, assume, perhaps, maybe, possibly, probably, likely, might, could, would, seem, appear, look, sound, feel, believe, suspect, guess, wonder, question, doubt, uncertain, unsure, unclear, confuse, puzzle, key, important, crucial, critical, essential, problem, issue, challenge, difficulty; 
 *Phrases:* let's consider, let's think, let's suppose, let's assume, it seems that, it appears that, the key is, the problem is, the issue is, I think, I believe, I suspect, I wonder |

## B   EXPERIMENTAL DETAILS

All experiments in this paper are conducted under carefully controlled and standardized settings. For the generation phase, we use the `vLLM` framework (v0.10.1). During the evaluation phase, all experiments are performed using the `transformers` library (v4.55.4). Prompt templates are adapted by making modifications to the default template provided with each model.

We evaluate three models from different families: Qwen3-8B (Yang et al., 2025), Llama-3.1-Nemotron-Nano-8B-v1 (Bercovich et al., 2025), and gpt-oss-20B (Agarwal et al., 2025). Other variants in these series, such as Llama-3.3-Nemotron-Super-49B-v1 and gpt-oss-120B, exceed the memory budget of a single 80GB GPU and therefore fall outside the range that can be served on our setup. Similarly, the smallest released models in some other families (*e.g.*, GLM-4.5-Air-110B (Zeng et al., 2025a) and KAT-V1-40B (Zhan et al., 2025)) are already larger than what fits within this limit. For different models, we use their officially recommended sampling settings:

- **Qwen3-8B:** we adopt the officially recommended configuration: for the thinking mode we set temperature to 0.6, Top P to 0.95, and Top K to 20; for the non-thinking mode we set temperature to 0.7, Top P to 0.8, and Top K to 20.

- **Llama-Nemotron-8B:** we use its official usage recommendations: we set temperature to 0.6, and Top P to 0.95 for the thinking mode and use greedy decoding for the non-thinking mode.

- **gpt-oss-20B:** we follow its official settings, using temperature = 1.0.

The forced-decoding experiments in Section 5.2 are conducted on five standard benchmarks: AIME24 (Li et al., 2024), AMC (Li et al., 2024), MATH500 (Hendrycks et al., 2021), Minerva Math (Lewkowycz et al., 2022), and OlympiadBench (He et al., 2024a). In the main text, other experiments are conducted on the AIME24 dataset (Li et al., 2024). Experiments on the other four datasets can be found in Appendix C. For all settings, each problem is sampled five times. All experiments are run on NVIDIA A100 GPUs with 80GB memory.

## C    MORE EXPERIMENTS ABOUT METACOGNITIVE KEYWORDS

We also conduct experiments on the outputs of Qwen3-8B on other datasets: AMC (Li et al., 2024), MATH500 (Hendrycks et al., 2021), Minerva Math (Lewkowycz et al., 2022), and Olympiad-Bench (He et al., 2024a). The results are shown in Figure 7. It can be found that other datasets share similar phenomena to the AIME24 dataset. For Self-Correction (B), Reasoning Articulation (C), and Deliberation (D), the thinking mode contains more metacognitive keywords, reflecting self-monitoring, explanation, and deliberation. For Planning & Structuring (A), it has more planning words overall but a lower frequency, as it spends more effort debating and verifying between steps, unlike the non-thinking mode that lists steps directly.

## D    USE OF LLMS

In preparing this manuscript, we used LLMs solely as a writing assistant to polish language and improve clarity of exposition. The research ideas, experimental design, implementation, analysis, and conclusions were entirely conceived and carried out by the authors. The LLMs did not contribute to research ideation, data analysis, or substantive scientific content. The authors take full responsibility for the final text.

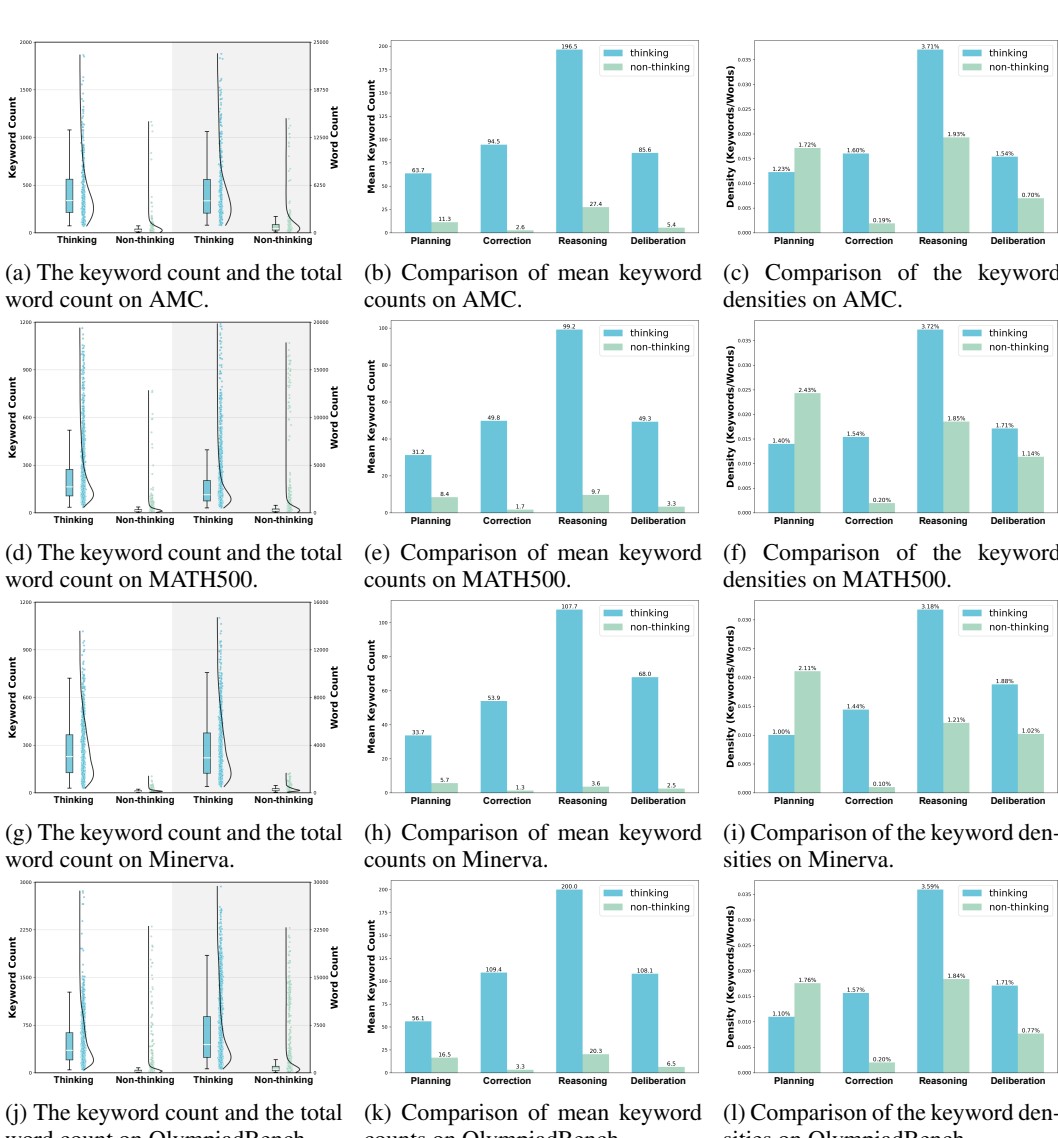

(a) The keyword count and the total word count on AMC.

(b) Comparison of mean keyword counts on AMC.

(c) Comparison of the keyword densities on AMC.

(d) The keyword count and the total word count on MATH500.

(e) Comparison of mean keyword counts on MATH500.

(f) Comparison of the keyword densities on MATH500.

(g) The keyword count and the total word count on Minerva.

(h) Comparison of mean keyword counts on Minerva.

(i) Comparison of the keyword densities on Minerva.

(j) The keyword count and the total word count on OlympiadBench.

(k) Comparison of mean keyword counts on OlympiadBench.

(l) Comparison of the keyword densities on OlympiadBench.

Figure 7: The statistics about the metacognitive keywords of Qwen3-8B on other datasets. Although the thinking mode has more metacognitive keywords among all four types, its density of planning keywords is less than that of the non-thinking mode.

