# OpenReview forum: "Anatomy of a Hybrid Mind: Deconstructing Hybrid Reasoning in Large Language Models"
_ICLR.cc/2026/Conference — ICLR 2026 Conference Withdrawn Submission_

### Official Review · Reviewer_8fmK · 2025-10-23

**Soundness:** 2
**Presentation:** 1
**Contribution:** 1
**Rating:** 4
**Confidence:** 4

**Summary:**

This paper conducts a mechanistic analysis of "hybrid reasoning" models, which can switch between a fast, intuitive (non-thinking) mode and a slow, deliberate (thinking) mode. The authors present three main findings: (1) that the compatibility of these modes is explained by their low token-level divergence (5-10% disagreement); (2) that the thinking mode is characterized by a structured "metacognitive protocol," identified via a taxonomy of keywords related to planning, self-correction, articulation, and deliberation; and (3) that the switch is governed by a localized, single-token trigger

**Strengths:**

See below

**Weaknesses:**

See below

**Questions:**

**Overall Rating:**
Weak Reject. The paper's core analysis of how metacognitive capacity differentiates reasoning modes is genuinely interesting and important. However, the work structurally overstates the significance of its other two main findings, which are largely unsurprising given prior knowledge of LLMs. Furthermore, the evidence provided does not yet establish a sufficiently strong causal link between the identified metacognitive protocol and the performance gains from the thinking mode. The paper needs substantial reframing to focus on its strongest contribution and to provide the necessary causal evidence to support its central claims.

### **Major Comments**

The following are things that, if adequately addressed, could increase my score.

1. **Refocus the Narrative and De-emphasize Unsurprising Findings.**
The paper should be rewritten to focus on its most valuable contribution—the analysis of metacognitive capacities. The claims of major surprisingness for the other two findings should be substantially toned down.

	- **Compatibility via Low Divergence (Section 3):** The finding that the same model weights can support both modes with >90% token agreement is not highly surprising. LLMs are known to handle multiple languages or domains within a single weight set, often with significant token-level disagreement - English vs French involves major token-level disagreement!
	- **Single-Token Switch (Section 5):** The existence of a localized trigger is also expected. We already know that model behavior can be controlled by a small number of tokens in a system prompt, that's how people tell it what mode to use. The discovery that a specific completion token also functions as a switch is not a given, but fairly unsurprising given this.
2. **The Metacognitive Analysis Needs Stronger Causal Evidence.** This is by far the most interesting part of the paper, but the current evidence is correlational. The results do not meaningfully distinguish whether these metacognitive keywords are a primary driver of reasoning performance or merely a stylistic artifact. To strengthen this, I suggest:

	- **Ablation Experiment:** A crucial experiment would be to prevent the model from generating these metacognitive keywords in thinking mode (e.g., by setting their logits to -∞) and measuring the effect. How much does reasoning performance degrade towards the non-thinking baseline? How much does the token-level agreement change? This would directly test the causal importance
	- **Baseline Comparison:** Analyze a standard base model (e.g., Llama) prompted with a simple "think step-by-step" instruction. This would help distinguish between general properties of chain-of-thought text versus specific behaviors learned during hybrid-reasoning fine-tuning.
3. **Address Methodological Confounding in Probabilistic Analysis.**
The analysis linking high NLL/KL divergence to specific tokens (e.g., after the <think> tag, or near keywords) is likely confounded by token entropy. Tokens at decision points or sentence starts are naturally higher-entropy, which would lead to lower log-probs from any model. To disentangle "surprise due to mode switch" from "predictable uncertainty," I recommend presenting a scatter plot of the generating model's token entropy vs. the cross-mode KL divergence to demonstrate that the effect is non-trivial.

### Minor Comments:
The following are unlikely to change my score, but are comments and suggestions that I hope will improve the paper.

- The observation that planning keyword density decreases in the "high" thinking mode while other metacognitive keywords increase is surprising to me, and interesting - naively, the 3 modes should change the reasoning duration but preserve the ratios of words. I'd be interested in seeing further exploration here

---

> ### Author Response · Authors · 2025-12-02
>
> We thank Reviewer 8fmK for the thoughtful and constructive feedback. The comments are extremely valuable, and we will revise the paper accordingly. In particular, we will incorporate the suggested experiments, including the ablation that suppresses metacognitive tokens in thinking mode, the baseline comparison with a standard model under a CoT prompt, and the scatter plot relating token entropy to cross-mode KL divergence. These insights are very helpful, and we appreciate the reviewer’s careful evaluation.

---

### Official Review · Reviewer_VLVT · 2025-10-29

**Soundness:** 2
**Presentation:** 3
**Contribution:** 2
**Rating:** 4
**Confidence:** 3

**Summary:**

This paper investigates three questions about hybrid reasoning in language models, where “hybrid reasoning” is defined as the co-existence of a deliberate reasoning mode and a non-thinking mode. The authors perform a series of experiments to study (1) why these two modes are compatible, (2) why the two modes lead to differences in behavior, and (3) how the activation of one mode is controlled.

**Strengths:**

The topic of hybrid reasoning models is timely and interesting. The visualization tool may also be useful to the research community. I also appreciate that the paper is also well-written and easy to follow.

**Weaknesses:**

I felt that several of the claims in this paper were too strong, and the conclusions were not adequately supported by the evidence. For example, with regards to Takeaway 1 (pg. 4), I would argue that having probability mass on mostly the same tokens is not a “key reason why” (l. 201) non-reasoning and reasoning modes can exist in the same architecture – it is merely a symptom of this co-existence. To truly explain *why* these two modes can coexist, the paper would need to show *why* the reasoning and non-reasoning modes share very similar token distributions. As is, I find Takeaway 1 too strong, and the results in Section 3 not very satisfying.

Another strong conclusion is when the authors find that the thinking mode of Qwen3-8B leads to outputs with a higher frequency of “metacognitive keywords”, which leads the authors to then write: “This confirms that the “thinking” process is characterized by an active protocol of self-monitoring, logical explanation, and careful consideration” (l. 286-288). I found this conclusion to be highly anthropomorphized, and does not follow from simply observing high counts of words like “Wait” or “Hold on”.

In Section 4.3, the authors also find that words like “Perhaps” or “Let’s check” tend to be followed by high-surprisal tokens under the evaluator model (if I’m understanding correctly). The authors go on to write: “This experiment not only quantifies our initial visual findings but, more importantly, directly links the linguistic-level concept of metacognitive keywords to the probabilistic-level uncertainty within the model, thus forging a complete explanatory chain.” (l. 308-310). Would a more direct test of this conclusion involve looking at the entropy after these kinds of trigger words *within* a reasoning mode, not looking at the NLL of the words under an evaluator model? I was a bit confused by how the findings in this section form a “complete explanatory chain”.

There were also key details missing from the description of the taxonomy (Section 4.2). How was this constructed? How were the keywords/phrases sorted into the categories? If this was all done manually, how was the taxonomy validated? How were the outputs of Qwen3-8B categorized (just naive string matching)?

**Questions:**

In lines 243-246, you write: “Conversely, the thinking mode is surprised by the non-thinking mode’s tendency to jump directly into calculations or final answers. These observations strongly suggest that the core difference is not in the model’s knowledge but in its application of strategic, metacognitive behaviors.” Could you explain what you mean by this? What, quantitatively, is this conclusion based on?

Could you add more discussion of the limitations of your work to the Discussion?

Please also see my question above about the taxonomy in Section 4.2.

---

> ### Author Response · Authors · 2025-12-02
>
> We thank Reviewer VLVT for the feedback. Below, we strictly clarify some misunderstandings and answer some of the questions.
>
> ---
> ***1. Misunderstanding of the Logic Behind Takeaway 1***
>
> Thank you for the comment. We would like to clarify that our argument in Section 3 follows a two-step structure that may have been misunderstood. The root cause for compatibility is established using two different models: base vs. fine-tuned models. We show that SFT/RL specialization changes <10% of tokens (Table 1). This small behavioral divergence explains why multiple behaviors can coexist within one model. Hybrid models are then used only as a verification, not as the causal explanation. Table 2 demonstrates that thinking and non-thinking modes follow the same low-divergence pattern, confirming that hybrid modes fit the same principle.
>
> Thus, we are not claiming that "overlapping token distributions" in hybrid models explain coexistence; rather, low divergence (established from fine-tuned models) is the underlying reason, and hybrid models simply exhibit this same property. We will revise the text to make this ordering clearer.
>
> ---
> ***2. How were the four metacognitive categories chosen? Do they count using direct string matching?***
>
> We thank the reviewer for the question. Our taxonomy is based on the **Metacognitive Regulation framework [1] in Cognitive Science**, which includes three main processes: planning, monitoring, and evaluating. Planning matches our Planning & Structuring category, and evaluating matches our Self-Correction category. During our analysis, we found that monitoring appeared far more often than the other processes and covered too many different kinds of behaviors. To make it more precise, we divided it into two clearer behaviors: Reasoning Articulation, which involves checking the logic of the answer, and Deliberation, which involves checking confidence in the answer. This combination of theory and data-driven refinement led to the four categories used in the paper. As shown in Appendix A, to support the taxonomy introduced in the main text, we collect the complete list of metacognitive keywords and phrases used in our analysis, and count them by direct string matching.
>
> [1] Metacognition and cognitive monitoring: A new area of cognitive-developmental inquiry. American psychologist 34.10 (1979): 906.

---

### Official Review · Reviewer_HDku · 2025-10-30

**Soundness:** 2
**Presentation:** 3
**Contribution:** 1
**Rating:** 4
**Confidence:** 4

**Summary:**

This paper offers a mechanistic investigation of hybrid reasoning in LLMs, systems that alternate between fast, intuitive responses (“non-thinking”) and slow, deliberate reasoning (“thinking”). It studies how these modes coexist within a single model, what distinguishes them behaviorally, and how they are internally controlled.
Empirically, the authors analyze open-source hybrid models such as Qwen3-8B, Llama-Nemotron-8B, and gpt-oss-20B, using a combination of cross-evaluation, token-level divergence analysis, and causal intervention.
They find that thinking and non-thinking modes diverge on less than 10% of tokens, that the “thinking” mode corresponds to a structured metacognitive protocol involving planning, self-correction, reasoning articulation, and deliberation and that in reasoning models with different modes (e.g., gpt-oss), reasoning intensity scales smoothly in a continuous spectrum.
The authors interpret these as evidence that hybrid reasoning arises from compact, local control signals modulating a stable underlying model.

**Strengths:**

Clear empirical framing: The paper is well-organized and methodically progresses from high-level compatibility to fine-grained causal mechanisms.
Careful empirical analysis: The cross-evaluation setup and forced-decoding interventions are clean and reproducible. They convincingly show that hybrid reasoning can be controlled by minimal prompt-level toggles.
Solid mechanistic focus: The combination of probabilistic divergence, NLL visualization, and causal probing is a decent methodological step toward interpretability of reasoning models.

**Weaknesses:**

The main issue I have with the paper is that contributions seem incremental and oversold. Most of the “surprising” claims are anticipated by prior/concurrent interpretability and reasoning-behavior papers, but usually mixed with more interesting follow-up insights in those papers (https://arxiv.org/abs/2506.01939, https://arxiv.org/abs/2510.07364, https://www.arxiv.org/abs/2503.01307, https://arxiv.org/abs/2506.18167). Especially the taxonomy of behaviors and the finding that the models only diverge on few tokens are not very surprising/novel in my opinion. The empirical finding that reasoning intensity is continuous or controlled by local tokens is interesting, and the main novelty.

**Questions:**

The paper claims a “localized, single-token control mechanism.” Could the authors provide stronger evidence that this effect isn’t just prompt-conditioning? For instance, do internal activations or attention maps actually differ systematically right after that token?

---

> ### Author Response · Authors · 2025-12-02
>
> We thank Reviewer HDku for the feedback.
>
> We will not submit a full rebuttal, but we would like to offer a brief clarification. Most of the works cited in the weaknesses section are either very recent preprints or papers that have just been accepted by NeurIPS (even one firstly uploaded to arXiv in October). Given their timing, we believe it is not entirely fair to expect comprehensive comparisons against all of them. We appreciate the reviewer’s perspective and will incorporate relevant discussions in future revisions.

---

### Official Review · Reviewer_M2Nr · 2025-11-01

**Soundness:** 2
**Presentation:** 3
**Contribution:** 2
**Rating:** 2
**Confidence:** 4

**Summary:**

This paper studies hybrid reasoning by analyzing LLM and LRM models in 3 different directions: First, it looks at how similar the token generation is between reasoning and non-reasoning modes. Second, it identifies important keywords that show when a model is engaging in deliberate reasoning. Third, it examines how the model switches between these modes and what triggers the thinking process.

**Strengths:**

The paper is clearly written and easy to follow. The motivation is reasonable and the topic, understanding hybrid reasoning in large language models, is timely and relevant. The authors provide a coherent structure and explain their ideas in a way that makes the main points easy to follow. The experiments are generally sound and cover a fair range of setups, and the results are presented in a clear way. Overall, while not overly ambitious, the paper makes a solid attempt to unpack how hybrid reasoning emerges in current models and raises questions that are worth exploring further.

**Weaknesses:**

Section 3 – Missing baseline and unclear interpretation of NLL and Agreement: There is no baseline to understand whether the reported Top-1 Agreement or Average NLL values are meaningful. For example, a 91.95% agreement might sound high, but it’s unclear how different that is from comparing any two models. Without a reference point, we can’t tell if this number really shows compatibility or just reflects general similarity between LLMs. Also, both metrics can be misleading because most tokens in a reasoning sequence are low-entropy and easy to predict. Prior work [1] shows that only a few "high-entropy" tokens actually drive the reasoning path. This means the average NLL is dominated by many predictable tokens with very low loss, making the results look more similar than they actually are. In fact, the lower Average NLL for texts generated by Reasoning models further supports this explanation, since its outputs are longer, the average is taken over more low-entropy tokens, which naturally lowers the NLL and reinforces the idea that these numbers mostly reflect token distribution bias rather than true behavioral alignment.

[1] Beyond the 80/20 Rule: High-Entropy Minority Tokens Drive Effective Reinforcement Learning for LLM Reasoning

Section 4.1 – Unclear interpretation of "surprising" tokens: There is a fundamental issue with the analysis in Section 4.1. The paper identifies "surprising" tokens based on high NLL values from the evaluation model, but it’s unclear whether these tokens are also surprising for the generation model itself. I suggest comparing the NLL difference between the two models. If the effect disappears, it would mean these tokens are generally important, not specific to the difference between reasoning and non-reasoning modes.


Section 4.2 – Issues with the taxonomy and analysis of metacognitive keywords: (1) Plots (a) and (b) in Figure 3 are misleading because reasoning models produce longer outputs, so the higher keyword counts are largely an artifact of sequence length. It would be more meaningful to report keyword ratios rather than raw counts; those two plots could be removed or replaced by normalized versions. (2) Plot (c) lacks error bars, and a statistical test (e.g., a t-test) should be included to verify that the observed differences are significant. This would strengthen the claim that the taxonomy actually distinguishes the two modes.

**Questions:**

Use of "fast" and "slow" thinking: Why are these terms used? Are they simply referring to the LLM and LRM modes? Also, since all the evaluated models are trained on reasoning-heavy data, how meaningful is the distinction between fast and slow thinking in this context?


Robustness of the metacognitive taxonomy: How were the four metacognitive categories chosen? Could there be more (or fewer) meaningful categories? Are these categories grounded in established findings from cognitive science, or were they defined empirically for this study?

---

> ### Author Response · Authors · 2025-12-02
>
> We thank Reviewer M2Nr for the feedback. Below, we strictly clarify some misunderstandings and answer some of the questions.
>
> ---
> ***1. Interpretation of NLL/Agreement & comparison to Wang et al. [1]***
>
> The reviewer suggests that our metrics are misleading based on the findings of Wang et al. [1]. However, this stems from a misunderstanding of the differing scopes between the two studies, which **we clarified in Footnote on Page 4**. Wang et al. analyze the entropy within a single reasoning mode, focusing on which tokens drive the generation internally. In contrast, our work compares the logits across two distinct modes (reasoning vs. non-reasoning). Our finding concerns the structural alignment between these modes. The high agreement demonstrates that the two modes are foundationally aligned, a conclusion that stands independent of the entropy distribution within a single trace.
>
> [1] Beyond the 80/20 Rule: High-Entropy Minority Tokens Drive Effective Reinforcement Learning for LLM Reasoning
>
> ---
> ***2. How were the four metacognitive categories chosen?***
>
> We thank the reviewer for the question. Our taxonomy is based on the **Metacognitive Regulation framework [1] in Cognitive Science**, which includes three main processes: planning, monitoring, and evaluating. Planning matches our Planning & Structuring category, and evaluating matches our Self-Correction category. During our analysis, we found that monitoring appeared far more often than the other processes and covered too many different kinds of behaviors. To make it more precise, we divided it into two clearer behaviors: Reasoning Articulation, which involves checking the logic of the answer, and Deliberation, which involves checking confidence in the answer. This combination of theory and data-driven refinement led to the four categories used in the paper.
>
> [1] Metacognition and cognitive monitoring: A new area of cognitive-developmental inquiry. American psychologist 34.10 (1979): 906.
>
> ---
> ***3. Clarification of the meaningfulness of "fast" vs "slow" thinking in LLMs***
>
> For clarity, we use “fast” and “slow” thinking to distinguish between heuristic, automatic prediction (System 1) and more deliberate, step-by-step reasoning (System 2). Even though the evaluated models are trained on reasoning-rich data, the *fast* mode remains meaningful because it enables the model to produce answers without generating lengthy reasoning traces. This provides users with an additional interaction option, allowing them to choose between efficiency and transparency depending on their needs.

---

### Note · Authors · 2025-12-02

I have read and agree with the venue's withdrawal policy on behalf of myself and my co-authors.